# Diversity and Host Interactions among Virulent and Temperate Baltic Sea *Flavobacterium* Phages

**DOI:** 10.3390/v12020158

**Published:** 2020-01-30

**Authors:** Emelie Nilsson, Oliver W. Bayfield, Daniel Lundin, Alfred A. Antson, Karin Holmfeldt

**Affiliations:** 1Centre for Ecology and Evolution in Microbial Model Systems (EEMiS), Department of Biology and Environmental Science, Faculty of Health and Life Sciences, Linnaeus University, SE-39231 Kalmar, Sweden; emelie.nilsson@lnu.se (E.N.); daniel.lundin@lnu.se (D.L.); 2York Structural Biology Laboratory, Department of Chemistry, University of York, York YO10 5DD, UK; oliver.bayfield@york.ac.uk (O.W.B.); fred.antson@york.ac.uk (A.A.A.)

**Keywords:** genomics, aquatic, lytic, lysogeny, host range

## Abstract

Viruses in aquatic environments play a key role in microbial population dynamics and nutrient cycling. In particular, bacteria of the phylum *Bacteriodetes* are known to participate in recycling algal blooms. Studies of phage–host interactions involving this phylum are hence important to understand the processes shaping bacterial and viral communities in the ocean as well as nutrient cycling. In this study, we isolated and sequenced three strains of flavobacteria—LMO6, LMO9, LMO8—and 38 virulent phages infecting them. These phages represent 15 species, occupying three novel genera. Additionally, one temperate phage was induced from LMO6 and was found to be competent at infecting LMO9. Functions could be predicted for a limited number of phage genes, mainly representing roles in DNA replication and virus particle formation. No metabolic genes were detected. While the phages isolated on LMO8 could infect all three bacterial strains, the LMO6 and LMO9 phages could not infect LMO8. Of the phages isolated on LMO9, several showed a host-derived reduced efficiency of plating on LMO6, potentially due to differences in DNA methyltransferase genes. Overall, these phage–host systems contribute novel genetic information to our sequence databases and present valuable tools for the study of both virulent and temperate phages.

## 1. Introduction

In the majority of aquatic environments, bacteriophages (phages) vastly outnumber their bacterial hosts [1] and have a significant impact on the ecosystem as a result [2]. The impact on the microbial community depends on a variety of factors, including the host range and efficiency of infection on different strains [3,4], the number and character of metabolic genes carried by the phage [5,6,7], the metabolic reprogramming of the host population [8,9,10], and the replication strategy of the phage [11,12]. Phages have been estimated to kill ~20% of the bacterial population in the ocean every day [2], thereby contributing to the available pool of nutrients for the microbial community and preventing their transport to higher trophic levels [13,14]. In addition to lysing their microbial hosts, temperate phages have the ability to integrate into the genome of their hosts and remain dormant until they are induced to undergo lytic replication. In this way, temperate phages stay protected during times of harsh conditions [15]. Temperate phages can also offer benefits to the host, such as homoimmunity that prevents further phage infections [11], toxin production [16], and changes to the host gene expression [17], all of which increases the fitness and competitiveness of the lysogenised bacteria. Whilst a large proportion of bacteria isolated to date appear to contain integrated prophages [18,19,20], the patterns of lysogenic replication in aquatic environments do not show clear patterns. Large seasonal variations in the lytic-lysogenic switch have been observed, and it has been suggested that lysogenic replication is favoured under conditions of low bacterial abundance and productivity [21,22,23].

Bacteria of the phylum *Bacteroidetes* inhabit a wide range of environments, ranging from Antarctic soil, to Sahara sand dust [24], and the human gut microbiome [25]. In aquatic environments, *Bacteriodetes* constitutes a large proportion of the microbial community [26,27], where they play important roles in phytoplankton bloom recycling [28] and degradation of biopolymers [29]. So far, relatively few phages infecting *Bacteroidetes* have been characterised, for example, compared to phages infecting *Gammaproteobacteria*, *Actinobacteria*, and *Firmicutes* [30], and hence there is little known about the virus–host interactions influencing this major member of the ocean microbial community. Among isolated *Bacteroidetes* phages, the majority infect members of *Flavobacteriaceae*—the largest aquatic family within *Bacteroidetes* [31]. These phages originate from various aquatic environments [32,33,34,35]. They include a large collection of *Cellulophaga baltica* phages [36,37], which contains phages related to the abundant human gut crAssphage [38,39,40]. The *Cellulophaga* phages have proved to be an important model system for investigations of phage–host interactions, ranging from host range variations to transcriptomic analyses [4,8,10,41,42,43]. *Bacteroidetes* phages have also shown their potential as bio-control agents in aquatic environments [44,45], and a large diversity of both virulent and temperate phages infecting *Flavobacterium psychrophilum* have been isolated from fish farms [46,47,48]. As *F. psychrophilum* is a fish pathogen that causes large economic losses [49], the potential of using phage therapy to control this disease is of great importance. Overall, these comprehensive phage collections constitute an ideal model system for both the investigation of genetic diversity and phage–host interactions [48,50,51]. 

Given the importance of *Bacteroidetes* and their phage–host interactions in aquatic environments, we aim to isolate new *Flavobacterium* strains and phages. By isolating bacteria of different strains within the same genus, we wanted to explore the diversity within phages infecting them as well as the interactions between these phages and the different bacterial isolates.

## 2. Materials and Methods

### 2.1. Bacterial Isolation and DNA Extraction

Water for bacterial isolation was collected at 2 m depth at the Linnaeus Microbial Observatory (LMO; N 56° 55.8540′, E 17° 3.6420), situated 10 km east off the coast of Öland, Sweden, with a Ruttner sampler on 16 September 2015. Bacteria within the water were grown by spreading 100 µL Baltic Sea water onto Zobell agar plates (1 g yeast extract (Becton, Dickinson and Company (BD), Franklin Lakes, NJ, USA), 5 g bacto-peptone (BD), and 15 g of bacto-agar (BD) in 800 mL filtered (Whatman glass microfiber filer, GF/C, GE Healthcare, Chicago, IL, USA) Baltic Sea water and 200 mL Milli-Q water: this, and all other media used, were autoclaved at 121 °C for 20 min), from where colonies were picked and clean-streaked three times. DNA extraction of the genomic DNA of bacteria LMO6, LMO8, and LMO9 was conducted using the E.Z.N.A. Tissue DNA kit (Omega bio-tek, Norcross, GA, USA) according to the manufacturer’s instructions. Briefly, cells were lysed, protease treated, and bound to a HiBind DNA Mini column. The bound DNA was washed with DNA wash buffer diluted with 100% ethanol before it was eluted in 10 mM Tris. The quality and quantity of the DNA were verified with Nanodrop (Thermo Fisher Scientific, Waltham, MA, USA) and Qubit (high sensitivity DNA kit; Invitrogen, Life Technologies, Carlsbad, CA, USA), respectively, and sequenced at SciLife/NGI (see Section 2.7 “Sequencing of Viral and Bacterial Genomes”).

### 2.2. Viral Isolation

Water for viral isolation was collected on 16 September 2015 at LMO. To remove bacteria and larger organisms, 10 L of water was prefiltered through 0.22 µm sterivex filters (Merck Millipore, Darmstad, Germany) before viral particles were concentrated using a 30 kDa tangential flow filter (Merck Millipore). The resulting 100–150 mL was further concentrated using a 50 kDa Amicon Ultra centrifugal filter (Merck Millipore) to 13–30 mL.

The viral enriched water was used in plaque assays with the three bacteria isolated in this study, LMO6, LMO8 and LMO9, as hosts to discover viral infection. Bacteria were grown overnight in liquid Zobell medium (1 g yeast extract (BD) and 5 g bacto-peptone (BD) in 800 mL filtered Baltic Sea water and 200 mL Milli-Q water) with gentle agitation. The bacteria were mixed with the viral enriched water and molten top-agar (marine sodium magnesium (MSM) buffer: 450 mM NaCl (Sigma, St Loius, MO, USA), 50 mM MgSO_4_ × 7H_2_O (Merck Millipore) and 50 mM Trizma base (Sigma), pH 8; with 0.5% low melting point agarose (Thermo Fisher Scientific)). The mixture was spread evenly onto Zobell agar plates. The plates were incubated at room temperature (RT) for two days and plaque formation was monitored on a daily basis. On two of the bacteria, LMO6 and LMO9, 30 plaques were originally isolated for each bacterium, while only the five plaques that were formed on LMO8 were isolated. Phage isolates were purified by picking individual plaques with a sterile 100 µL pipette tip. The end of the pipette tip (containing the phages) was thereafter submerged in MSM to disperse the phages. The phages were then re-plated as described above and the procedure was repeated three times. Pure isolates were harvested by adding 5 mL MSM to fully lysed plates, the top agar-layer was shredded with an inoculation loop, and the plate incubated on a shaking table (40 rpm) for at least 30 min. The phages suspended in MSM were collected into a falcon tube, centrifuged for 10 min at 10,000× *g*, filtered through 0.2 µm syringe filter (BD) and stored at 4 °C.

### 2.3. Host Range Investigation

In order to determine the host range of the isolated phages, we used the three bacteria, LMO6, LMO8 and LMO9, as well as four other bacteria that shared 16S rRNA gene similarity with the previous three; Flavobacteriaceae bacterium BAL304 (accession: KM586862), Flavobacteriaceae bacterium BAL314 (accession: KM586887), Flavobacteriaceae bacterium BAL330 (accession: KM586875) and Flavobacteriaceae bacterium BAL346 (accession: KM586918). In order to produce plaque assays, they were grown in MLB (0.5 g Casein hydrolysate (Merck Millipore), 0.5 g bacto-peptone (BD), 0.5 g yeast extract (BD), 3 mL glycerol (Sigma) in 800 mL filtered Baltic Sea water and 200 mL Milli-Q water) overnight. They were grown in MLB instead of Zobell medium since the bacteria aggregated in Zobell medium but stayed more dispersed in MLB. Overnight culture (300 µL) was mixed with top-agar (3.5 mL) and dispersed on an agar plate, then a 100 µL pipette tip was dipped in the viral isolate and a line was drawn across the prepared agar plate according to the molten streaking for singles method [52]. This was repeated three times, after which a new pipette tip was used to draw three new lines intersecting the previous lines in an angle; this was repeated so that four sets of lines were drawn. When this method provided clear plaques, standard plaque assays were performed in order to count the efficiency of plating. The efficiency of plating was also performed with all phages on their original host of isolation. Plates were incubated for 24–48 h. The results were visualised with Tidyverse (version 1.2.1) [53] in R (version 3.5.1) [54] using RStudio (version 1.1.383) [55].

### 2.4. Prophage Induction Experiments with LMO6

LMO6 was grown in MLB until exponential growth was established, as measured with an optical density at 600 nm with a CO8000 cell density meter (WPA, Cambridge, UK). Thereafter, 25 mL of LMO6 was dispersed into a sterile petri dish (covering the bottom to approximately 1 mm depth) and subjected to UV irradiation with wavelength of 254 nm for 10 s by a handheld UV-lamp (Model UVGL-58 mineral light lamp, 6W, Upland, CA, USA) at a distance of 15 cm without the lid of the petri dish. In addition, 5 mL of LMO6 was dispersed in a similar manner, but not UV-irradiated, to act as control. After the treatments, the bacteria were diluted 10x in fresh MLB in sterile falcon tubes and kept in the dark with agitation for five hours and OD was measured every hour. After five hours, 2 mL of bacteria was filtered through a 0.22 µm syringe filter (Merck Millipore) to collect the supernatant containing free viral particles. The flow-through was used to enumerate viral particles by plaque assays, as previously described, with LMO6, LMO8 and LMO9 as hosts. The plates were kept up to 36 h in order to verify plaque formation. To obtain enriched stocks of only the temperate phage, flow-through from the non-UV treated LMO6 was used to perform plaque assays on LMO9. Here, the flow-through was diluted in a 10-times dilution series and fully lysed plates were harvested as described above. This was repeated until a high-titre phage stock was received.

### 2.5. Morphological Examination Using Transmission Electron Microscopy (TEM)

TEM was conducted on high-titre phage lysate. Carbon-formvar coated copper grids (Agar Scientific Ltd., Stansted, UK) were plasma cleaned in a PELCO easiGlow for 60 s at 0.38 mbar (air) and 20 mA. Lysate (5 µL) was applied to a grid for 60 s then removed by wicking. The grid was washed with deionised water (5 µL) and stained with 2% *w/v* uranyl acetate (5 µL (Agar Scientific Ltd.)). Grids were imaged using an FEI Tecnai 12 G2 BioTWIN microscope with tungsten filament operating at 120 kV. Phage particles were measured using ImageJ (version 1.52a).

### 2.6. Viral DNA Extraction

DNA extraction from the phages was performed with the Wizard PCR DNA Purification Resin and Minicolumns (Promega, Madison, WI, USA). First, newly harvested, high-titre phages were added to 1 mL resin and thoroughly mixed before being pushed through a minicolumn. Second, the column was washed twice with 1 mL 80% isopropanol and residues of isopropanol were removed by centrifuging the column for 2 min at 10,000× *g*. Last, to elude the DNA, pre-heated (80 °C) 10 mM Tris (Trizma base (Sigma), pH 8) was added, the column was vortexed and then centrifuged at 10,000× *g* for 30 s. The DNA was verified with regards to quality and quantity with Nanodrop (Thermo Fisher Scientific) and Qubit (high sensitivity DNA kit; Invitrogen, Life Technologies), respectively.

### 2.7. Sequencing of Viral and Bacterial Genomes

Library preparation for the extracted DNA from both bacteria and phages was done with Nextera XT (Illumina Inc., San Diego, CA, USA) and sequencing was performed by SciLifeLab/NGI (Solna, Sweden) on a HiSeq 2500 (Illumina Inc.), which resulted in paired-end 125 bp reads (average 1.2 million reads per sample).

### 2.8. Assembly, Annotation, and Pan-Genome Analysis

Raw reads were trimmed to remove adapters and poor quality reads (Trimmomatic version 0.30, settings: -PE –threads 2 -phred33 ILLUMINACLIP:nextera_linkers.txt:2:30:10 LEADING:3 TRAILING:3 SLIDINGWINDOW:4:15 MINLEN:30) [56]; quality was evaluated with FastQC [57]. Trimmed reads were then assembled into contigs with Spades (version 3.6.0, settings: --careful –t 8 --pe1-1 --pe1-2 -o) [58] and Abyss (version 1.3.5, settings: -np 16 –k61 –q3 –coverage-hist –s –o) [59], and the assembler providing best quality was chosen (Appendix A). CheckM was used to assess bacterial genome completeness [60]. Bacterial genomes were annotated with NCBI’s Prokaryotic Genome Annotation Pipeline (PGAP) [61] while open reading frames (ORFs) in the phage genomes were called with Phanotate (version 1.0.1; default settings) [62] and manually inspected. Viral ORFs were then annotated with Diamond (version 0.9.10, settings: blastp -p 5 -q -k 100 -e 0.001 --sensitive --tmpdir -f 6 -o) [63] against the NCBI non-redundant database (3 May 2019) and hmmsearch (version 3.2.1, settings: --tblout -E 1e-3 --cpu 1) [64] against the Pfam database (24 February 2017) [65]. A significant alignment had an e-value of less than 0.001. Core- and pan-genomes were calculated with Roary (version 3.6.2, settings: -s -i 70) [66]. The comparison of phage genomes based on nucleotide identity was performed with Gegenees (version 3.0.0, settings: fragment size of 500 and step size of 500, blastn) [67], which fragments the genomes and compares identity across the entire genomes. The similarities were then plotted with Tidyverse (version 1.2.1) [53] with R (version 3.5.1) [54] using RStudio (version 1.1.383) [55]. Bacterial genomes were compared with Pyani (version 0.2.7, settings: average_nucleotide_identity.py -m ANIb) [68], and run through PHASTER [69] to identify potential prophage regions.

### 2.9. Phylogenetic Analysis

GTDBtk (version 0.3.2) [70] was used to place the bacterial genomes of LMO6, LMO8 and LMO9 in the Genome Taxonomy Database [71] tree, which was conducted through analysis of a set of marker genes, and calculate shared average nucleotide identity with the reference genomes. The tree was visualised with Dendroscope (version 3.5.9) [72].

To perform phylogenetic analysis of the phage genomes, a set of reference genomes were selected based on the criteria that they should share at least one gene with significant amino acid similarity (e-value less than 0.001) to any of the phages in this study. For the lytic phages, only other viral genomes were selected, while bacterial contigs with several genes with significant amino acid similarity to the temperate phage were also selected. The protein sequences of this reference set, containing 37 genomes, were then acquired through NCBI and phylogenetic analysis of the references and LMO-phages were conducted through VICTOR [73]. For bacterial genomes with potential prophage regions with similarity to the temperate phage, only the relevant, potential prophage region was used for the VICTOR analysis. VICTOR was performed using pairwise comparisons of the nucleotide sequences, with settings recommended for phages [73], through the Genome-BLAST Distance Phylogeny (GBDP) method [74]. The distances were used with FastME to infer a tree with branch support [75] for the D0 formula. Also, VICTOR estimated taxon boundaries (species, genus, family) with the OPTSIL program [76] with recommended clustering thresholds [73].

### 2.10. Mapping of Environmental Reads to the Phage Genomes and Host Bacteria

Reads from viral metagenomes from LMO, sampled between 2012 and 2015 (BioProject accession number: PRJNA474405) [77], were mapped against the type phages for each genus (mumin9-1, pippi8-1, tant8-1, laban6-1) with bowtie2 (v2.3.5.1, settings: --ignore-quals –mp = 1,1 –np = 1 –rdg = 0,1 –rfg = 0,1 --score-min = L,0,-0.1) [78]. For a phage to be counted as present in the dataset, the genome needed to be covered by at least 75% of its length by reads with at least 90% sequence identity [79], where the first part was calculated with SAMtools [80] and BEDtools [81] and the second was acquired through the bowtie2 settings. Viral metagenomes were also gathered from publicly available datasets (Global Ocean Sampling (GOS): CAM_P_0000914, CAM_P_0000915, CAM_P_0001109, and Tara Ocean Viromes: ERR594353-4, ERR594356, ERR594358-9, ERR594361-2, ERR594364-5, ERR594368-70, ERR594374, ERR594376, ERR594378, ERR594384, ERR594391-3, ERR594395-6, ERR594398, ERR594400-1, ERR594403-4, ERR594406, ERR594410, ERR594412) and recruited against the genomes in the same manner, except for data from the GOS dataset where unpaired data were mapped (setting -f). 

Reads from bacterial metagenomes from LMO [82] were mapped against the three bacterial strains LMO6, LMO8, and LMO9 (settings as above, except: --score-min = L,0,-0.03 for species identity, --score-min = L,0,-0.2 for genus identity). To assign a read to a particular bacterial strain, the read had to share at least 97% identity to the reference genome, while 80% identity was required to assign the read to a bacterial genus, and 60% of the genome should be covered. Also, the 16S rRNA genes from LMO6, LMO8 and LMO9 were compared with 16S amplicon sequencing data from LMO [77] (blastn) and the counts for the 100% identical amplicon sequencing variant (ASV) were retrieved from the dataset.

### 2.11. Mapping of Reads to Confirm Prophage Induction

Mapping of reads was performed in order to confirm that the temperate phage was induced prior to DNA extraction and sequencing, and not only sequenced as an integrated prophage within the bacterial genome. During phage DNA extraction, there might occur contamination from bacterial DNA, which, after sequencing, could result in low abundances of bacterial host sequences. Therefore, reads from one sequencing sample (well number 103 that contained tooticki6-1 and laban6-1, the lytic and temperate phage identified after sequencing, respectively) were mapped with bowtie2 (as above) against the lytic phage genome (tooticki6-1) and the bacterial contig of LMO6 that contained the prophage (contig name: NODE_12_length_95344_cov_81.199_ID_11796, specific accession number: WIBH01000012.1). Coverage and depth were calculated as described above. The mapping against the bacterial contig was plotted with tidyverse with R (version 3.5.1) [54] using RStudio (version 1.1.383) [55].

### 2.12. Accession Numbers

All sequences in this study are deposited in bioproject PRJNA566314 at DDBJ/ENA/GenBank.

Bacterial strains, assembled as high-quality draft genomes, were deposited with the specific accession numbers WIBH00000000 for *Flavobacterium* sp. LMO6, WIBI00000000 for *Flavobacterium* sp. LMO9 and WIBJ00000000 for *Flavobacterium* sp. LMO8.

Viral genomes were deposited under the accession numbers MN812203–MN812241.

Reads from well 103 were deposited under the accession number SAMN13638723. 

## 3. Results

### 3.1. Characteristics of Flavobacterium spp. Strains

Three bacterial strains (LMO6, LMO8, and LMO9) were isolated from the Baltic Sea in September 2015. All three had similar colony morphology, with a yellow to orange colour. LMO8 had colonies that were easily retrieved from the agar plates, whereas both LMO6 and LMO9 colonies were sticky and difficult to pick from the agar plates. While the 16S rRNA genes for the three strains were 99.7–100% identical at the nucleotide level, LMO6 and LMO9 shared more than 99% average nucleotide identity (ANI) to each other across 98.4–98.9% of the genomes while LMO8 shared 85% ANI with the other two across 60–64% of the genomes. Genome comparison and placement in the reference tree showed that all strains were placed within *Flavobacterium* and were most similar to *Flavobacterium sp000169355* (Figure 1): LMO6 and LMO9 with 96.8% ANI and LMO8 with 86.1% ANI. The bacterial isolates are therefore named *Flavobacterium* sp. strain LMO6, *Flavobacterium* sp. strain LMO8 and *Flavobacterium* sp. strain LMO9. Of the three bacterial strains isolated in this study, no prophage region was detected in LMO8, an incomplete prophage region of 8.9 kb was detected in LMO9, while LMO6 had an incomplete prophage region of 37.3 kb.

### 3.2. Genomic Characteristics of Lytic Phages

Of the 65 phages that were originally picked for isolation on the three bacterial strains, 38 lytic phages could be purified and sequenced. All sequenced phages had double-stranded DNA genomes and identical k-mers at both ends of the genomes, indicating that they were circular. The lytic phages’ genomes ranged between 34,627 and 40,542 nucleotides, and they contained between 53 and 76 predicted genes (Appendix A). The majority of the phages (34) were isolated from LMO6 or LMO9 and shared >50% average nucleotide identity (ANI) to each other. The remaining four phages were isolated with LMO8 as host and three were highly similar (>90% ANI) while the last phage shared no similarity to the others. With a genus cut-off of more than 50% ANI [83] across all genomes and a species cut-off of more than 95% ANI, these newly isolated phages are placed in three new genera and 15 new species based on all-versus-all genome comparisons, as well as VICTOR analysis (Figure 2, Appendix A). These newly isolated phages were not sufficiently similar to any reference genome to be able to establish further phylogenetic relationships (Figure 3). The three genera are proposed to be named Muminvirus, Pippivirus and Tantvirus based on the type species for each genus, Flavobacterium virus Mumin, Flavobacterium virus Pippi and Flavobacterium virus Tant. The isolates are named according to the ICTV’s recommendations [84], e.g., Flavobacterium phage vB_FspS_mumin9-1, where 9 stands for the bacterial host of isolation (LMO9) and 1 is the phage isolate number.

Genes belonging to the phage in Tantvirus did not share any similarities with either Mumin- or Pippivirus genes and Muminvirus and Pippivirus shared only four genes with low (27–29%) amino acid similarity (e-value less than 0.001; Appendix A). Thus, the pan-genomes (based on more than 70% amino acid similarity) were calculated within each genus. The pan-genome of Muminvirus consisted of 142 genes, including 26 genes in the core-genome, and the pan-genome of Pippivirus consisted of 56 genes, including 50 genes in the core-genome. Tantvirus only consisted of one phage isolate with 62 genes. Within the pan-genome of Muminvirus, 66 genes had significant hits (e-value less than 0.001) to the NCBI nr database but only 37 could be assigned a function (Appendix A). The pan-genome of Pippivirus had a slightly higher ratio of hits to the NCBI nr database (33 hits, 59%, e-value less than 0.001) and 13 genes could be assigned a function (Appendix A). For Tantvirus, 39 of the 62 predicted genes had hits to the NCBI nr database and 14 of those were assigned a function (Appendix A). Despite the dissimilarities between the genera, the annotations of the genes all belong to the same categories, generally even the same predicted functions. These were mainly functions involved in DNA processing, replication and recombination, as well as structural proteins and proteins involved in DNA packaging (Figure 4).

### 3.3. Genomic Characteristics of the Temperate Phage

During the sequencing of the lytic phages, two complete genomes were assembled from pure phage cultures isolated using LMO6 as host. Comparison to the sequenced phages from LMO9 and the sequenced genome of LMO6 concluded that one of the phage genomes was from a phage with lytic replication (described above), tooticki6-1. The extra phage genome was detected in the sequence of the bacterium LMO6 and overlapped the incomplete 37.3 kb prophage identified by PHASTER. Therefore, this proved to be a temperate phage integrated into the genome of LMO6. The complete temperate phage genome could be assembled from nine of the sequenced viral isolates from LMO6 and these sequences were identical at the nucleotide level. This temperate phage was named Flavobacterium phage vB_FspS_laban6-1. Mapping of reads from the sequenced sample, including both tooticki6-1 and laban6-1, resulted in high coverage to the lytic phage (1680 read coverage), while the bacterial contig containing laban6-1 only was covered to a larger extent (50-260 read depth) over the region where the temperate phage was integrated (Figure 5).

The temperate phage genome was 43,236 nucleotides long, and 57 potential genes could be predicted. The genome was double-stranded DNA and determined to be circular as both ends of the sequence had identical k-mers. Three of the predicted genes had sequence similarity to genes within Muminvirus, but showed only 25–37% amino acid identity (e-value less than 0.001; Appendix A). Of the 57 genes, 44 showed amino acid similarity to proteins within databases but only 12 could be assigned to a function (Figure 4, Appendix A). As for the lytic phages, functions involved in DNA processing, replication and recombination as well as structural proteins and proteins involved in DNA packaging were identified, as well as a putative tyrosine site-specific recombinase involved in integration (Figure 4). VICTOR analysis using nucleotide similarities placed the temperate phage as a distinct species within the same genus of both previously isolated phages (e.g., *Flavobacterium* phage 6H [47]) and potentially integrated prophages (e.g., *Flavobacterium psychrophilum JIP02/86* [47]; Figure 3). However, Gegenees analysis based on nucleotide identity across the entire genomes (blastn analysis) did not report any similarity between the temperate phage and the references. Therefore, we propose that our new temperate phage should be placed in its own genus. The proposed name for this phage genus is Labanvirus, based on the type species Flavobacterium virus Laban, which is named after the isolated phage.

### 3.4. Morphology

In electron micrographs, it was observed that the type phage for Tantvirus, tant8-1, exhibited a siphovirus morphology, with a capsid diameter of 68 nm (standard deviation, ±1 nm, n = 13) and a tail length of 160 nm (±7 nm, n = 13; Figure 6a). The type phage for Pippivirus, pippi8-1, exhibited a myovirus morphology, with a capsid diameter of 63 nm (±2 nm, n = 9) and a tail length of 90 nm (±2 nm, n = 9; Figure 6b). mumin9-1, the type phage for Muminvirus, exhibited a siphovirus morphology, with only a slight curvature of the tail (Figure 6c), with a capsid diameter of 65 nm (±2 nm, n = 30) and a tail length of 153 nm (±4 nm, n = 30). Phage laban6-1 exhibited a siphovirus morphology, with a capsid of 68 nm (±2 nm, n = 12) and a tail length of 215 nm (±7 nm, n = 12; Figure 6d). In electron micrographs of the tooticki6-1 lysate samples, which should contain both the lytic phage tooticki6-1 and the temperate phage of LMO6, laban6-1, two different phage particles types with siphovirus morphology were indeed observed. Phage tooticki6-1 (Figure 6e(i)) had a head diameter of 64 nm (±2 nm, n = 15) and a tail length of 127 nm (±4 nm, n = 15), which was similar in appearance and lengths to mumin9-1 (compare Figure 6c to Figure 6e(i)). Present alongside were relatively few phage particles possessing heads of diameter 65 nm (n = 1), and longer, flexible tails of 223 nm length (n = 1), characteristic of isolates of laban6-1 (compare Figure 6d to Figure 6e(ii)).

### 3.5. Host Range

The host ranges of the different phage species varied between the different genera (Figure 7 and Appendix A). The type phages of the species pippi8-1, lotta8-1 and tant8-1, which were isolated on LMO8, could infect both LMO6 and LMO9 but with much lower efficiency of plating compared to their efficiency on their original host. None of the phages within Muminvirus were able to form plaques on LMO8, but they could infect both LMO6 and LMO9 independent of which of the two bacterial strains that were used for isolation. However, while all phages isolated with LMO6 as their original host had an equally high efficiency of plating on both LMO6 and LMO9, some of the phages isolated with LMO9 had a three to four orders of magnitude lower efficiency of plating on LMO6 (Figure 7 and Appendix A). To investigate if the last infected host could influence this behaviour, morran9-1 and lillamy9-1 were grown on LMO6 before new plaque assays were performed on LMO6 and LMO9. This resulted in similar efficiency of plating of morran9-1 and lillamy9-1 on both hosts: morran9-1 produced 2.5 × 10^8^ plaque-forming units (PFU) ml^−1^ on LMO6 and 3.9 × 10^8^ PFU ml^−1^ on LMO9, and lillamy9-1 produced 1.0 × 10^8^ PFU ml^−1^ on LMO6 and 2.2 × 10^8^ PFU ml^−1^ on LMO9.

### 3.6. Induction of laban6-1 From LMO6

During the induction experiment, the optical density of LMO6 and the UV-irradiated LMO6 continued to increase throughout the incubation (Appendix A). Free, induced laban6-1 phages could be detected through plaque-formation on the lawn of LMO9, while no plaques were formed on LMO6 or LMO8. Free phages originating from bacteria-free filtrates from both the UV-radiated (1.3 × 10^8^ ± 1.1 × 10^8^ PFU ml^−1^, n = 3) and control LMO6 cultures (2.0 × 10^7^ ± 6.0 × 10^6^ PFU ml^−1^, n = 3) could be detected and the UV-radiated phages resulted in 3 to 6 times more plaques compared to the control.

### 3.7. Temporal and Spatial Variation

Mapping of viral metagenomic reads against the type phage of each genus resulted in the detection (>75% of the genome covered with reads [79]) of mumin9-1 in three LMO samples, August and September in 2013 and December in 2014 (Appendix A). None of the other phages showed great enough genome coverage to be regarded as present in any of the LMO metagenomes and none of the phages were detected in any of the GOS or Tara ocean viral metagenomes (Appendix A). Read-mapping of bacterial metagenomic reads from LMO showed that the bacterial isolates were not detected at species (97% read identity to the reference) or genus level (80% read identity to the reference) (Appendix A). Even though the three bacterial strains did not have 100% identical 16S rRNA genes, they all match a 448-nucleotide long 16S V3V4 amplicon sequence variant (ASV) in the LMO amplicon sequencing data with 100% nucleotide identity. This ASV was detected in summer samples (May–July) at LMO in 2011–2013 and 2015 (Appendix A).

## 4. Discussion

The phages described in this study belong to four different genera whose members infect three strains of *Flavobacterium.* The proposed names for these phage genera are Muminvirus, Pippivirus, Tantvirus and Labanvirus (Appendix A). Despite their genetic (Figure 2; Figure 3) and morphological differences (Figure 6), they were quite similar in genome size, ranging between 34 and 43 kb (Appendix A). This limited range of genome sizes is more uniform than the range of genome sizes seen in e.g., the *Cellulophaga* phages, which range in genome size from 6 and 145 kb [37], or the mycobacteriophages, which range between 40 and 160 kb [86]. Considering the large number of isolated phages for the LMO6 and LMO9 host strains, it could be suggested that the natural diversity of phages, with lytic replication that infect these strains, was low at the time of sampling, as only members of one genus could be isolated. However, the intra-genus diversity was relatively large with twelve different species isolated from the same timepoint. A single bacterial species maintaining several phage species of the same genus has been seen before, e.g., for the *Cellulophaga* phages, where two genera were isolated from the same seawater sample [36] and consisted of either four or three different species (phi13:2, phi18:3, phi 19:3, and phi 46:3 in the first genus, and phi12:1, phi17:1, and phi18:1 in the second; Figure 3, Appendix A, and [37]). In contrast, while only managing to isolate five, and sequence four, different phages for LMO8, these were divided into both different genera and had different morphologies (Figure 6a,b; Appendix A). These results likely reflect the diversity of *Flavobacterium* spp. phages at LMO at the time of sampling, and the number of plaques originally produced on the different hosts would suggest a higher abundance of muminphages, originally isolated on LMO6 or LMO9, compared to pippi- or tantphages, originally isolated on LMO8. While there is no sequenced viral metagenome for the time when the phages were isolated, mumin9-1 could be detected in viral metagenomes from previous years (Appendix A). This is also an indication that muminphages are more commonly abundant than the pippi- or tantphages. Since LMO8 supports phages that belong to different genera and have different morphologies, the competition of these phages for their host might prevent any of them reaching an abundance that can be detected in the viral metagenomes. The presence of muminphages did not coincide with the presence of the detected ASV that was similar to the host strains (Appendix A). Coinciding phage–host patterns have previously been seen at LMO for the barbaphages, a recently reported phage genus with myovirus morphology infecting a *Gammaproteobacteria* host, isolated from the same sample as the phages reported in this study [77]. The lack of coinciding temporal variations for the *Flavobacterium* host-strains and their phages reported here implies that there might be other, undescribed phages for the LMO6, LMO8, and LMO9 hosts that are prevalent in the Baltic Sea. Thus, further, high-resolution time series investigations of phage–host systems are called for.

About half, or more, of the genes for Mumin- (46%)-, Pippi (59%)-, and Tantvirus (62%) had similarities to previously known sequences, and the majority of best matches were to either various phages or to bacteria that belong to the same family as the bacterial host strains in this study, *Flavobacteriaceae*. Still, there was low similarity to any specific phage, which was confirmed by VICTOR analysis (Figure 3). The closest relationship to a previously known phage was a distant connection between tant8-1 and *Flavobacterium* phage Fpv5 (Figure 3), a similar-sized (42.6 kb) siphovirus isolated from fish farms in Denmark [46,48]. This points towards a shared ancestor of this viral type among phages infecting various *Flavobacterium* hosts in the Baltic Sea region. Besides this, no other close relationships were seen to previously isolated phages, even though phages infecting members of *Flavobacteriaceae* isolated from the Baltic Sea are relatively well-represented in culture collections [87], including the diverse *Cellulophaga* phage–host collection [37] and sea ice *Flavobacterium* phage 1/32 [88] (Figure 3).

Only 21–38% of the predicted genes in the different genera could be assigned a function (Appendix A). Even though the genes between the different genera had low or non-existing similarities, the functions of the genes in each genus are classified in the same categories. As often seen among phages, these functions are involved in (*i*) DNA processing, replication and recombination, and (*ii*) structural and packaging (Figure 4). Among the structural genes, we were able to identify putative terminase genes in tant-, mumin-, and pippiphages, and putative major capsid proteins in tant- and muminphages. These are among the genes that are considered as viral hallmark genes [89,90], and even though hallmark genes are expected to exist in all phages, they are not always correctly annotated since their identification relies on having similarities to previously sequenced and properly annotated genes. The lack of identified hallmark genes has previously been noted for other aquatic phages infecting *Flavobacteriaceae* [37]. While aquatic phages in particular are known to have genes involved in various metabolic processes, e.g., DNA metabolism [9,91], no metabolic genes were detected in any of the phages. Further, no genes involved in lysogenic replication were detected, indicating that mumin-, pippi-, and tantphages are only capable of lytic replication. This is also true for *Flavobacterium* phage Fpv5, which is related to tant8-1, where no genes that would indicate a possible lysogenic lifestyle were detected [48]. 

Phage laban6-1 is a prophage in LMO6 and was sequenced as a result of induction, which was confirmed by read mapping (Figure 5). If the phage had been assembled by sequences of contaminating bacterial DNA, the coverage would have been even across the entire bacterial contig. That laban6-1 is indeed a temperate phage capable of lytic replication and not only a cryptic prophage—a prophage that has lost the ability to produce virions due to e.g., genome rearrangement [12,92]—was further confirmed by plaque assays on the closely related host, LMO9. Moreover, laban6-1 was induced from LMO6 grown in liquid medium without any additional stressors, producing high numbers of laban6-1 particles, around tens of millions. However, the induction did not affect the host’s generation time (91 min, ±9), which was similar to that of LMO9 (92 min, ±21). Induction of temperate phages without any experimentally added stress is called spontaneous prophage induction (SPI) [93] and has been demonstrated previously [94]. SPI might give a bacterial host fitness advantages, for example, by promoting biofilm formation by the release of extracellular DNA or releasing toxins that increase the infectivity of non-induced bacteria (see [95] and references therein). Although none of these benefits from SPI were noted, LMO6 did not seem to be adversely affected by the induction, at least not considering its generation time. While none of the predicted genes in laban6-1 indicated any obvious benefit for the host (Appendix A), the failure of laban6-1 to produce plaques on LMO6 indicates that the presence of laban6-1 as a prophage in LMO6 resulted in resistance to new infections of laban6-1. This resistance could be caused by superinfection exclusion, i.e., prevention of similar phages entering the infected cell [96], or homoimmunity, i.e., resistance towards similar phage DNA within the cell [11]. Besides SPI, the increased phage titre in the UV-light exposure experiments accompanied by reduced bacterial OD (Appendix A) indicates that UV can act as an inducing agent for laban6-1. However, it should be noted that the UV-light that LMO6 was exposed to in the experiment had a wavelength of 254 nm, i.e., UVC, which does not pass through the ozone layer and that solar radiation usually does not cause virus induction in all lysogenised bacteria [97]. Thus, it is unclear if natural UV can be a cause for laban6-1 induction, which therefore calls for future investigations. While SPI is clearly occurring in LMO6 cultures at room temperature (22 °C), laban6-1 could not be detected infecting LMO9 at the time of sampling, which indicates that there were no free laban6-1 in the water used for phage isolation. However, temperature increases are a known inducing agent for prophages [98] and given the lower temperature (16.2 °C) at the time of sampling compared to room temperature (22 °C), this temperature increase might lead to the induction levels we detect. While not investigated further in this study, this opens up for additional questions of the effect of global temperature increases of the viral dynamics in aquatic environments and what effects these switches from lysogenic to lytic replication will have on microbial communities and nutrient cycling.

Of the predicted genes within laban6-1, 77% (44/57 genes) had hits to the queried databases, but only 21% (12 genes) could be assigned to a function. The large amount of hypothetical hits to bacterial genomes is likely a result of laban6-1 being a temperate phage and therefore sharing a lot of similarity with un-annotated prophages. The relatedness to sequenced prophages was confirmed by the VICTOR analysis, where laban6-1 is placed within the same genus as several potential prophage regions in *Flavobacterium* strains as well as potentially temperate *Flavobacterium* phages (Figure 3, Appendix A). These phages and bacteria have been isolated from various aquatic environments, including fish cultures (the phages and *Flavobacterium psychrophilum JIP02/86*), the freshwater Han River in Korea (*Flavobacterium* sp. CJ74), and mining wastewaters (*Flavobacteriales bacterium 32-34-25*). Thus, despite that laban6-1 was not detected in LMO or other viral metagenomes, this group of related phages is clearly prevalent in a large range of aquatic habitats and might have consequences for, e.g., economic interests through their infection of fish pathogens [47].

The host range of the *Flavobacterium* phages in this study was determined using efficiency of plating as a measure of successful propagation. The phages belonging to the Pippi- and Tantviruses, originally isolated on LMO8, could, besides infecting LMO8, also replicate on both LMO6 and LMO9; however, with a reduced efficiency of plating (Figure 7). This could be an indication that LMO6 and LMO9 are sub-optimal hosts for the Pippi- and Tantviruses, potentially by reducing the infection through defense mechanisms. These phages, which are able to infect two seemingly different species of bacteria, could be assumed to be more prevalent in the environment since they are able to infect several hosts [99]. However, this is contradicted by the mapping of viral metagenomes against tant8-1 and pippi8-1, which did not result in enough coverage to be able to count them as present at any time point. In contrast, the muminphages that only produced plaques on LMO6 and LMO9 (Figure 7) were detected in the viral metagenomes (Appendix A), potentially due to a specialisation on their host. Phages specialised on a specific host have been seen to result in larger burst sizes compared to more generalist phages infecting multiple hosts [43], which could explain why muminphages were more abundant in the viral metagenomes than the phages isolated on LMO8. While previous studies have reported that myoviruses have broader host ranges than siphoviruses [100], such patterns were not observed here. Instead, the siphovirus (tant8-1) and the myoviruses (pippiphages) isolated on LMO8 could infect the same host strains (Figure 7).

While all phages in Muminvirus could infect both LMO6 and LMO9, they did so with varying efficiency of plating. Phages originally isolated on LMO6 had equal efficiency of plating on both hosts, while some of the phages originally isolated on LMO9 had lower efficiency of plating on LMO6 (Figure 7). Such results have previously been seen among other aquatic phages, e.g., *Lightbulbvirus*, and the variation in the efficiency of infection has been shown to depend on the last host that was infected [4]. This also appears to be the case for the muminphages, as the LMO9 phages with reduced efficiency of plating on LMO6 increased their efficiency after replication on LMO6. This shift in efficiency of plating depending on the latest infected host is commonly associated with the restriction-modification system and methylation or glycosylation of the phage genome [96,101,102]. Phages are also able to provide their own methylation to evade host restriction enzymes. While the methylation patterns of the LMO6 and LMO9 phages have not been investigated, all LMO9 phages which originally had reduced efficiency of plating on LMO6 lacked a specific methyltransferase that the LMO9 phages with high efficiency of infection on LMO6 possessed (Appendix A). Thus, it is likely that some of the Muminvirus species have the ability to methylate their genomes in a way that provides full protection against the LMO6 and LMO9 bacteria restriction enzymes, while others require additional host-derived methylation to be able to reach a high efficiency of plating on LMO6 [96,103].

The LMO *Flavobacterium* phage–host systems described here contribute to an ever-growing resource of phage–host model systems for the study of aquatic virology. The absence of metabolic genes in the phage genomes analysed, for example, those involved in nucleotide metabolism, was unexpected, given their recent detection in many aquatic phage genomes. Bacterium LMO6 was found to be infected by virulent phages as well as harbouring a UVC-inducible temperate phage. Such phage–host systems appear to be quite rare among aquatic environmental isolates. Hence, the characterised LMO6–laban6-1 system represents a unique opportunity for future investigations focused on the effects of different environmental conditions on lytic and lysogenic infection cycles.

## Figures and Tables

**Figure 1 viruses-12-00158-f001:**
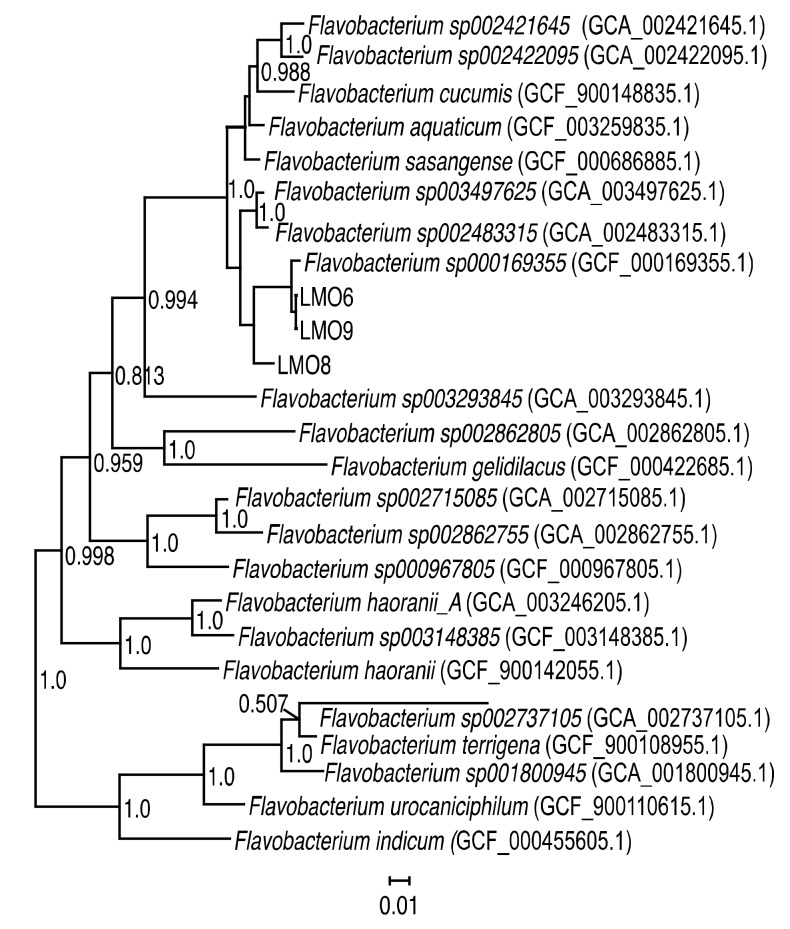
Phylogenomic placement of the three strains from this study—LMO6, LMO8, and LMO9—in the GTDB phylogeny [71]. The three strains were placed with GTDBtk [70] and here visualised together with their closest *Flavobacterium* spp. relatives. Numbers at nodes are bootstrap values (values below 0.5 are not shown).

**Figure 2 viruses-12-00158-f002:**
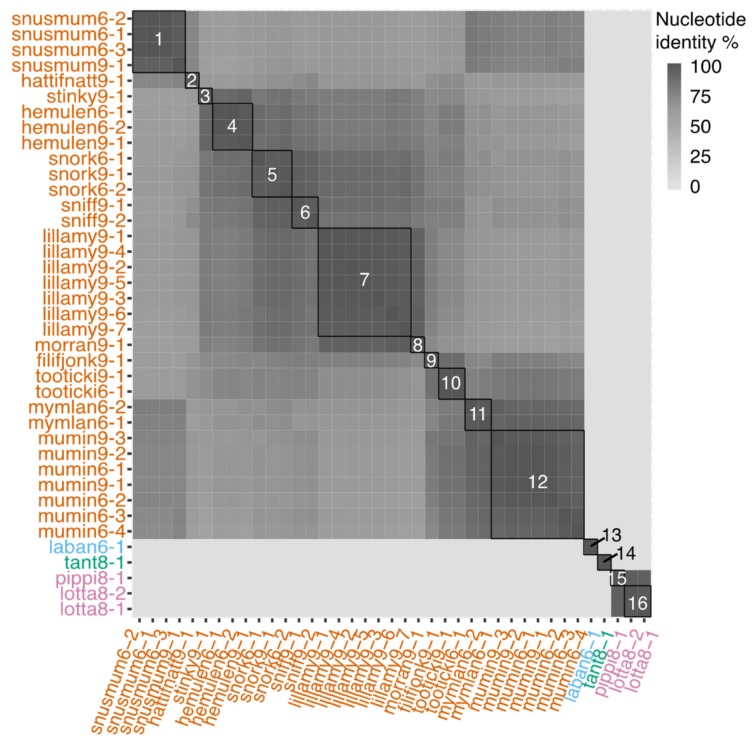
Similarity between the 38 lytic phages and the temperate phage based on Gegenees, which does a fragmented blastn between the entire genomes [67]; for details see Appendix A. Species are boxed with black while genera are coloured in the labels. Phages belonging to Muminvirus are orange, the isolate that constitutes Labanvirus is blue, Tantvirus is green and the isolates that belong to Pippivirus are pink. The species are named with the Flavobacterium virus and then 1 = Snusmum, 2 = Hattifnatt, 3 = Stinky, 4 = Hemulen, 5 = Snork, 6 = Sniff, 7 = Lillamy, 8 = Morran, 9 = Filifjonk, 10 = Tooticki, 11 = Mymlan, 12 = Mumin, 13 = Laban, 14 = Tant, 15 = Pippi, 16 = Lotta.

**Figure 3 viruses-12-00158-f003:**
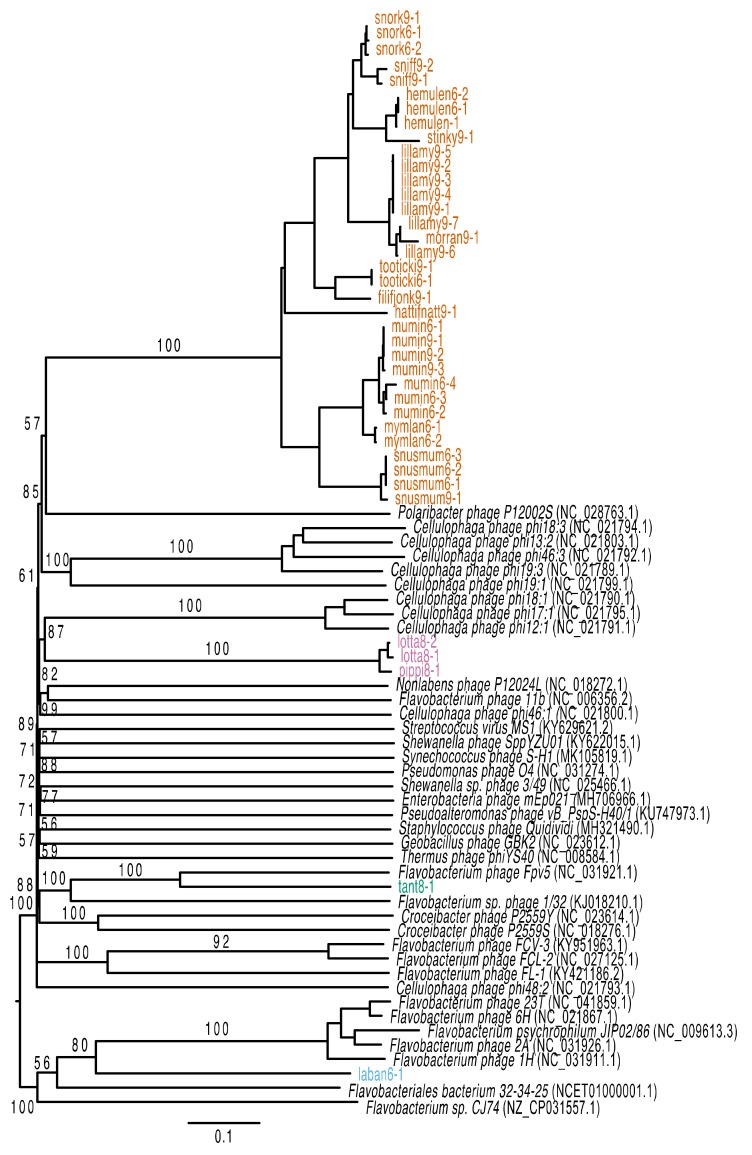
VICTOR tree produced with FastME based on nucleotide identity calculated with the D0-formula [73], with the newly isolated phages and reference genomes that share gene similarity to the novel phages presented in this study (accession number in parenthesis). The phages from this study are coloured based on the genera, as described in Figure 2. Bootstrap values above 50 are shown.

**Figure 4 viruses-12-00158-f004:**
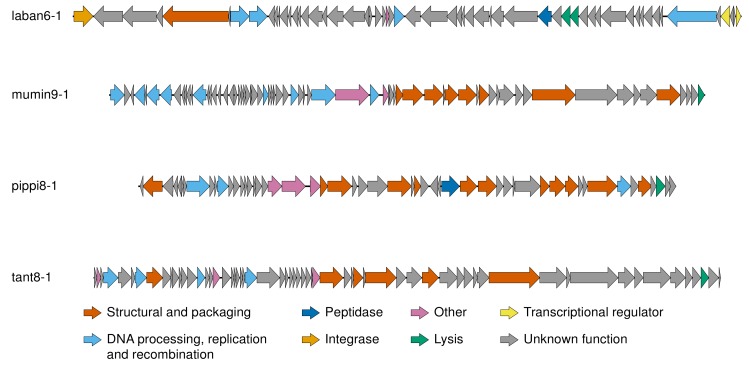
Genomic organisation of the type phages for each genus described in this study. Figure created with EasyFig [85]; for detailed information regarding the genomes and genes, see Appendix A. The genomes are displayed linearly for visualisation.

**Figure 5 viruses-12-00158-f005:**
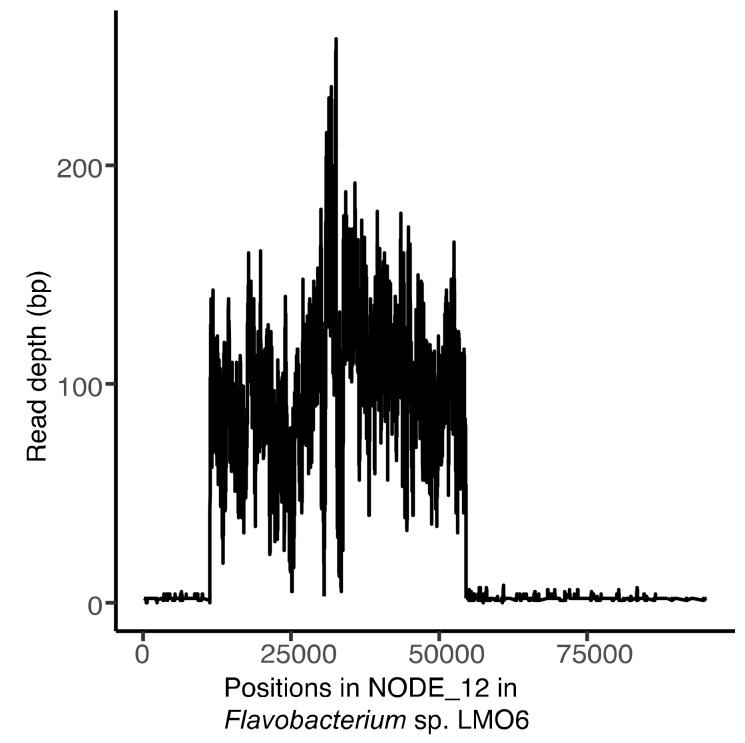
Coverage of viral reads (from well 103) against the bacterial contig that contained the prophage, laban6-1 (contig “NODE_12”).

**Figure 6 viruses-12-00158-f006:**
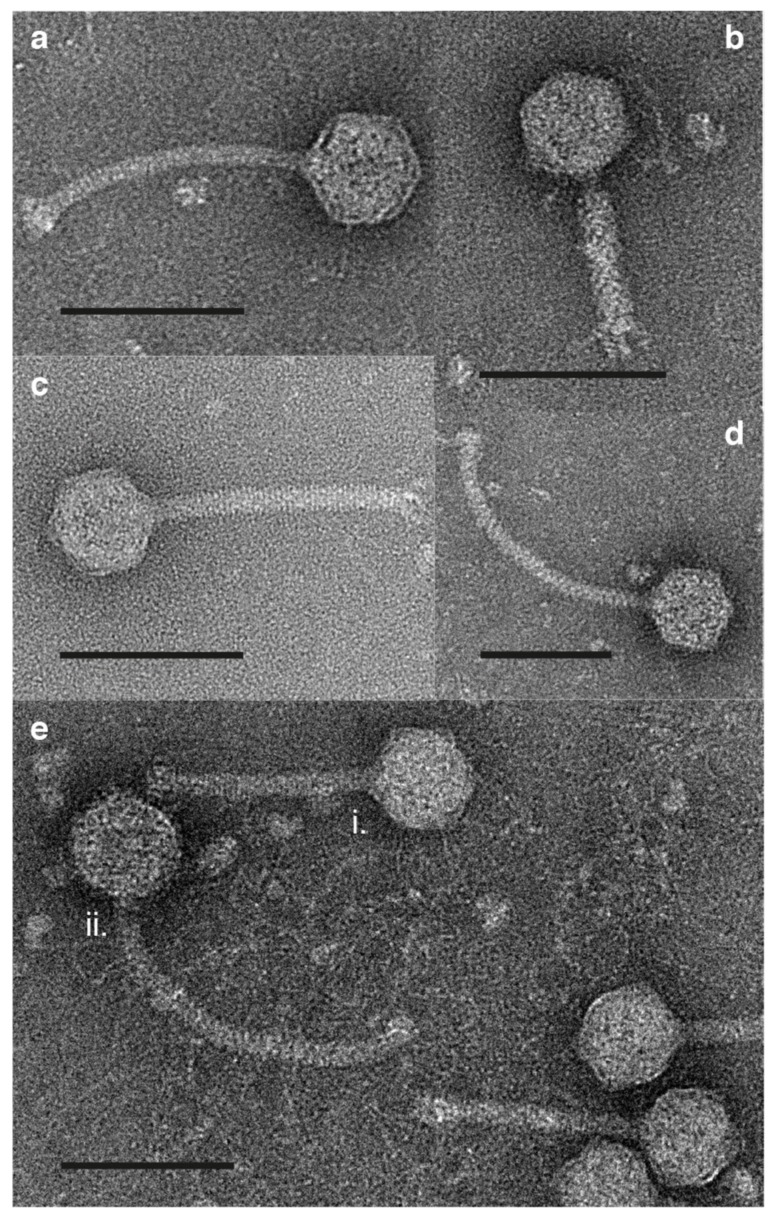
Negatively stained transmission electron micrographs of phage particles. (**a**) tant8-1, **(b**) pippi8-1, (**c**) mumin9-1, (**d**) laban6-1, (**e**) (i) tooticki6-1 and (**e**) (ii) laban6-1. Scale bars are 100 nm.

**Figure 7 viruses-12-00158-f007:**
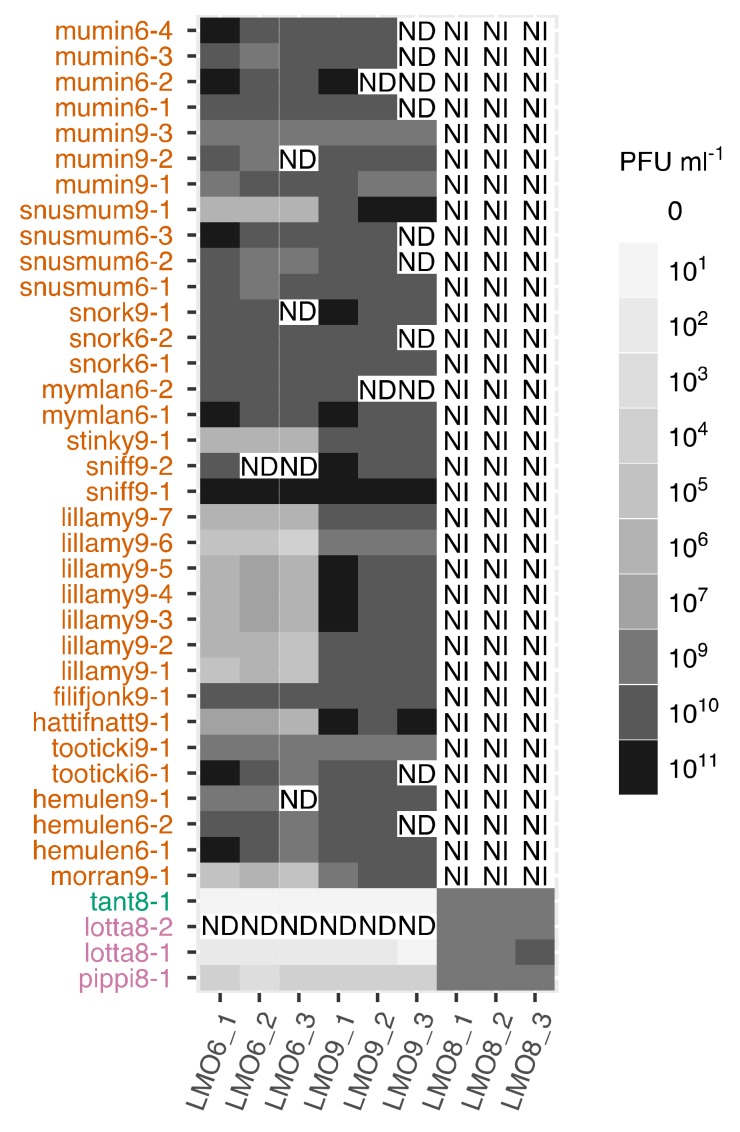
Host range tests for the type phages within each species with the three bacterial strains used within this study. ND = not determined, i.e., when the plaque assay has not been made; NI = no infection, i.e., when a plaque assay was performed but no plaques were detected. Phages are coloured according to genus, as in Figure 2. The number after, e.g., “mumin” indicates which bacterial strain the phage was originally isolated on, namely, 6 for LMO6, 8 for LMO8 and 9 for LMO9. The names on the *x*-axis are which bacterial strain the efficiency of plating was performed on, as well as if it was the first, second or third replicate (_1, _2, or _3).

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
