# Peer review of "Diversity and Host Interactions among Virulent and Temperate Baltic Sea Flavobacterium Phages"

_viruses, 2020, doi:10.3390/v12020158_

Round 1
Reviewer 1 Report
In this manuscript, Nillson and co-authors describe the isolation of three bacterial isolates and 39 phages, together with their phylogenetic and genomic characterization, and environmental distribution in the Baltic See. Overall, the manuscript is well written, although sometimes the phrasing could be improved and the color use in the figures should be checked. Generally, the methods used are adequate, and the conclusions reached match the results.
Specific comments below:
Materials and methods section: I would recommend to structure this sections into sub-sections, by using headings. For example, it took me a while to understand to what the paragraph at Lines 124-138 refers. A heading, e.g, “Prophage induction from LMO6” would’ve have made reading so much easier.
Line 77 and Line 113: how was the Baltic Sea water filtered? How was the sterility of the medium ensured, that is how were potential contaminations from the seawater (either by phages or bacteria) prevented? Was the medium autoclaved?
Line 101: Rephrase „with a sterile 100 l pipette tip that were dispersed in MSM”.
Line 136: You state that the flow-through from the non-UV treated LMO6 was used to “obtain enriched stocks of only the temperate phage”. Could you please elaborate? Why only the non-UV samples could be used, why not the UV-treated as well?
Line 137: It is unclear what “and the plate that was mostly lysed was harvested” means. Is it referring to dilution series?
Line 169: Please report the parameters used for Gegenees. Was there a threshold used to remove non-conserved regions? If yes, which one. If no threshold was used, did you check that, when comparing phage A against phage B, and phage B against phage A, the reported similarity values are not very different? If they are different, please mention it in the manuscript. According with the authors of Gegenees, if no threshold is used: “comparing two genomes of different size can sometimes give large differences in values depending on whether the small one is compared to the large one or the other way around.” On the other hand, if thresholds are used (which I do not recommend), than reported similarity values will represent only the conserved part of the genomes.
Line 203: what are tooticki6-1 and laban6-1?
Line 249: You are using a 50% threshold for genus demarcation, citing a 2017 paper. In my last communications with ICTV members, which took place last October, the genus threshold for Caudovirales was 70% average nucleotide identity. I would recommend that you confirm this threshold with ICTV members. If it is indeed 70%, use it to delineate your phage genera in this paper. Also, be aware that VICTOR phylogenetic assignment into genera/subfamilies, does not always matches the current recommendations of ICTV. The thresholds used by VICTOR are quite lower compared with those currently used by ICTV. I recommend checking first the thresholds you used with ICTV. Furthermore, I would recommend to check also the genera names with the ICTV members, to make sure they don’t match any other viral taxa. It would be a pity to publish this paper with some phylogenetic assignments, only to have them changed in a later taxonomic proposal by ICTV.
Figure 2: please provide the ANI values as a table in the SI. And eventually change the colors in the hittable, from gray based values to colors. It is rather difficult to understand the ANI values using the given gray value scale. Furthermore, the legend mentions that “Phages belonging to Muminvirus are black”, but I can’t see any black phage name.
Figure 3: check the coloring scheme for the phage names (see the comment about Figure 2 legend)
Line 271: The genes within Tantvirus did not share, I would rephrase to “Genes belonging to phages in the Tantivirus did not share …”
Section 3.2: Are the phage genomes complete? If yes, how was the genome completeness determined? For example, did the phage contigs obtained after assembly contain repeats at the end, indicating that the genomes can be circularized? Have the genome ends been determined, e.g. using PhageTerm software? Do the start and end positions in the genomic maps (Figure 4) correspond to the genomic ends, as determined with PhageTerm, for example?
Line 283: Comma needed after “Mainly”?
Line 324: Should “Muminvirus, mumin9-1, exhibited a siphovirus” be rephrased to “Mumin9-1, the phage type for Mumivirus, …..”?
Line 480: replace “has” with “have”
Author Response
In this manuscript, Nillson and co-authors describe the isolation of three bacterial isolates and 39 phages, together with their phylogenetic and genomic characterization, and environmental distribution in the Baltic See. Overall, the manuscript is well written, although sometimes the phrasing could be improved and the color use in the figures should be checked. Generally, the methods used are adequate, and the conclusions reached match the results.
Specific comments below:
Materials and methods section: I would recommend to structure this sections into sub-sections, by using headings. For example, it took me a while to understand to what the paragraph at Lines 124-138 refers. A heading, e.g, “Prophage induction from LMO6” would’ve have made reading so much easier.
Response: Thank you for your comment, we agree that this makes the methods more organised and easier to read and have now added subheadings.
Line 77 and Line 113: how was the Baltic Sea water filtered? How was the sterility of the medium ensured, that is how were potential contaminations from the seawater (either by phages or bacteria) prevented? Was the medium autoclaved?
Response: Thank you for your comment. The filtration was only conducted to remove larger particles and was done with Whatman glass microfiber filters, GF/C, which now has been added to the text, see lines 273-274. We have now clarified that all media used were autoclaved in order to ensure sterility, see lines 274-275.
Please note, all the line numbers we are referring to are the line numbers as they appear in the Microsoft Word track changes-version that we are submitting. Since the document contains tracking the numbers seem higher than they should, but we hope that they can make it easier to find where we have done the relevant revisions.
Line 101: Rephrase „with a sterile 100 l pipette tip that were dispersed in MSM”.
Response: Thank you for pointing out this phrasing. To increase clarity, we changed the sentence to: “… with a sterile 100 µl pipette tip. The end of the pipette tip (containing the phages) was thereafter submerged in MSM to disperse the phages. The phages were then re-plated as described above and the procedure was repeated three times, see lines 343-345.
Line 136: You state that the flow-through from the non-UV treated LMO6 was used to “obtain enriched stocks of only the temperate phage”. Could you please elaborate? Why only the non-UV samples could be used, why not the UV-treated as well?
Response: We chose the non-UV treated flow-through as this method is most similar to the way we prepared the viral stocks that were sent for sequencing, which is where we retrieved the first information of the presence of the temperate phage. When the phages were prepared for sequencing, the temperate phage was spontaneously induced, and thus, the non-UV treated/spontaneously induced sample made most sense to be used here.
Line 137: It is unclear what “and the plate that was mostly lysed was harvested” means. Is it referring to dilution series?
Response: Thank you for pointing this out. We have now clarified this sentence accordingly: “. Here, the flow-through was diluted in a 10-times dilution series and fully lysed plates were harvested as described above” line 382.
Line 169: Please report the parameters used for Gegenees. Was there a threshold used to remove non-conserved regions? If yes, which one. If no threshold was used, did you check that, when comparing phage A against phage B, and phage B against phage A, the reported similarity values are not very different? If they are different, please mention it in the manuscript. According with the authors of Gegenees, if no threshold is used: “comparing two genomes of different size can sometimes give large differences in values depending on whether the small one is compared to the large one or the other way around.” On the other hand, if thresholds are used (which I do not recommend), than reported similarity values will represent only the conserved part of the genomes.
Response: We apologies for not including the parameters directly, this was an error on our part. They are now included, see line 427. We did not use any threshold for the analysis, and the values are not very different no matter if we look at phage A against B or B against A, which can be seen in figure 2 where the values for each comparison are reported as a heatmap (e.g. both mumin9-1 against tooticki6-1 and tooticki6-1 against mumin9-1) and in Supplemental table S3 where each similarity value is reported.
Line 203: what are tooticki6-1 and laban6-1?
Response: tooticki6-1 is the lytic phage that was the intended target for the isolation and sequencing and laban6-1 is the temperate phage that was spontaneous induced and therefore isolated at the same time as tooticki6-1. This has been clarified see lines 478-479.
Line 249: You are using a 50% threshold for genus demarcation, citing a 2017 paper. In my last communications with ICTV members, which took place last October, the genus threshold for Caudovirales was 70% average nucleotide identity. I would recommend that you confirm this threshold with ICTV members. If it is indeed 70%, use it to delineate your phage genera in this paper. Also, be aware that VICTOR phylogenetic assignment into genera/subfamilies, does not always matches the current recommendations of ICTV. The thresholds used by VICTOR are quite lower compared with those currently used by ICTV. I recommend checking first the thresholds you used with ICTV. Furthermore, I would recommend to check also the genera names with the ICTV members, to make sure they don’t match any other viral taxa. It would be a pity to publish this paper with some phylogenetic assignments, only to have them changed in a later taxonomic proposal by ICTV.
Response: Thank you for your comment. As we all know, phage taxonomy is currently in an exciting state when known truth are changing fast. As you suggested, we have been in contact with Evelien Adriaenssens at ICTV. She confirms that 70% is the cut-off for assigning a genus where we would be quite sure that all members actually will be part of the same genus. However, given the genetic variation among viruses, Evelien also point out that different cut-offs can be needed for different phages (it is important that you point out what you use), and that it’s a cohesive group that defines a genus, more than only a similarity cut-off. According to Evelien, 50% can be used to investigate if the members belong to the same genus and since we did not find any reference genomes that shares 50% nucleotide identity (or even more than 20%) to any of the members in our suggested genera, we believe that this is an indication that they form a cohesive group, and therefore a genus.
For our dataset, shifting from a genus cut off of 50% to a genus cut off of 70% would have an effect on the muminphages, where we would not be able to create cohesive groups: e.g. some phages of a certain genus would share >70% identity with members of another genus while other members would not. Given this, as well as other important characteristics, e.g. similar in length, GC-content and gene content, we find that the use of a 50% genus cut off is the most suitable for the phages in our study.
Regarding the genera name, we have checked with ICTV and searched for the proposed names on NCBI as well as gone through all current proposals that are publicly available on ICTV and have not found any other names that would make ours valid. We thank you for encouraging us to investigate this.
Figure 2: please provide the ANI values as a table in the SI. And eventually change the colors in the hittable, from gray based values to colors. It is rather difficult to understand the ANI values using the given gray value scale. Furthermore, the legend mentions that “Phages belonging to Muminvirus are black”, but I can’t see any black phage name.
Response: The ANI values are now provided in Table S3. We understand that it is hard to distinguish between the different shades of grey, however after testing different colour scales we do not believe that any other colour display the differences in a better way. We have also tested different colours for each range, however we felt that this did not improve the figure either and therefore we kept the grey scale. Thank you for pointing out that we erroneously wrote that mumin were in black, it should be in orange and this is now modified, see line 588.
Figure 3: check the coloring scheme for the phage names (see the comment about Figure 2 legend)
Response: This is now fixed, see the response regarding figure 2.
Line 271: The genes within Tantvirus did not share, I would rephrase to “Genes belonging to phages in the Tantivirus did not share …”
Response: Thank you for this suggestion, we have modified the text with your rephrasing, see line 599.
Section 3.2: Are the phage genomes complete? If yes, how was the genome completeness determined? For example, did the phage contigs obtained after assembly contain repeats at the end, indicating that the genomes can be circularized? Have the genome ends been determined, e.g. using PhageTerm software? Do the start and end positions in the genomic maps (Figure 4) correspond to the genomic ends, as determined with PhageTerm, for example?
Response: The phage genomes are complete and circular based on identical k-mers, this has been clarified see lines 569-570. We have not used PhageTerm since that software is incompatible with the way our libraries were prepared before the Illumina sequencing. The start/end of the genomes have been decided based on an assumption that the structural genes should be at the end of the sequence since these are the last to be transcribed.
Line 283: Comma needed after “Mainly”?
Response: Thank you for your suggestion. The sentence has been rewritten accordingly: “These were mainly functions…” see line 618.
Line 324: Should “Muminvirus, mumin9-1, exhibited a siphovirus” be rephrased to “Mumin9-1, the phage type for Mumivirus, …..”?
Response: Thank you for this comment, we have rephrased according to your suggestion, see line 677.
Line 480: replace “has” with “have”
Response: Thank you for pointing this out, this has now been changed, see line 1110.
Reviewer 2 Report
Nielson et al have isolated and studied a number of phages infecting Flavobacterium strains from Baltic sea and analysed their genome sequencing. I was delighted to learn that they have decided to call these phages by using lovely character names originating from famous Finnish and Swedish writers Tove Jansson and Astrid Lindgren.
Although the study has been done with scientific rigors, the text is occasionally difficult to follow and understand. The manuscript text needs to be improved, especially the materials&methods and discussion. The figure legend and table titles and table column heading are occasionally incomplete.The work is quite descriptive. It would benefit if the discussion could link the study better to the known literature. Especially, I would like to know for example that is this kind of studies done before in Baltic and or related environments. Is the host range linked to the virus morphology observed? Myoviruses typically have broader host range than siphoviruses. The host genome sequencing is not documented so that at least I could not follow has it done here or somewhere else. Data and/or references need to be provided.
The virus taxonomy is moving forward quite fast. It would be good to use siphoviruses and myoviruses as a virus morphology in the text and forget Siphoviridae and Myoviridae (the taxonomic units), since families and morphologies does not always follow among members of the order Caudovirales. There are currently 10 families in the order Caudovirales
Line 2: Reconsider the title. Functionality does not necessarily describe the content of the work properly.
Genetic diversity and virus-host interactions among...
14: Family - refers to taxonomy? Group could be better
16: How many sequenced? This should be said in the abstract
21-24: Rephrase. Too detailed for abstract, make a conclusion.
53: infect members of Flavobacteriaceae
72: Materials and Methods: Please provide subheadings for this section.
78: MQ -> distilled water, correct also in other places
124-138: How you know that you have induced only one phage? Did you plaque purified it?
147: DNA extraction of phages is here, but genome extraction of the strains is described above. It is difficult to follow what has been done. Maybe separate the phage sequences and strain sequences, or otherwise clearly indicate what has been done. Use subheadings.
177: Please specify what is high
177: protein sequence similarity, see also below
180: how many?
200-207: mapping of ?? Please specify, difficult to understand
230: This does not say are the bacterial genome sequences whole genomes or draft genomes?
233: "Whole genome comparison" Where is the data of the whole genome sequences?
234: Flavobacterium genus and species names in italics, correct in all places
239: Please specify "the three strains" in the legend; add Acc Nos for each isolates into the figure; Is the figure based on whole genome sequence or on 16S?
242: lytic phages-> virulent phages: do you have data supporting this?
virulent phages have typically lytic life-cycle and lyse the cell in the end of its life-cycle, but can have also non-lytic exit strategy. The plaque can be seen e.g. due to the host growth retardation.
244: Are the genomes dsDNA? Circular or linear?
250: Do you mean "in three new genera and 15 new species"?
261: Muminvirus is orange
268: with the newly isolated phages (Delete lytic and temperate); Acc. Nos for the reference genomes would be goof to have in the fig. Are these phages all isolated and cultivable or does this include also prophages?
272: Known / Predicted functions for these shared genes?
274: What is the size range of the genomes within one genus?
287: add genome lengths, ORF / gene numbers above the arrows, mark the four shared genes (see comment above); Are these linear molecules?
290: phage cultures (you have not purified phage particles biochemically)
292: "a phage with lytic replication (described above), which was the phage that was supposed to be sequenced" (delete) -> concluded that one of the genomes was the phage (give here the name of the phage).
296: temperate phages -> sequences
298-302: Unclear. Rephrase
304: Have you used all obtained viral reads? What is NODE-12?
305: The temperate phage was 43,236 nucleotides long -> The temperate phage genome was 43,236 nucleotides long. Is this genome dsDNA, circular, linear?
306: What these genes are? Give gene names and refer to data in a Table.
310: a putative tyrosine...
313: Is Fig2 correct figure for this sentence? Add references as examples that what these similar phages are
328: the phages tooticki6-1 and laban6-1
330: Correct the Figure name
332-334: Is the culture mixture of several viruses?
336: Provide a high-resolution figure, now too low-resolution
342: Mark the original host of each virus in fig 7.
348: "on LMO6 and then harvested before new plaque assays were performed on LMO6 and LMO9."
-> on LMO6 and plaque assays were performed on LMO6 and LMO9.
349: plaques per ml -> PFU/mL; Correct in all places
347-351: Summarize what is the results. Influence?
353: What mean _1, _2, _3 in each bacterial isolates? Clarify this in the figure legend.
359: Have you checked that the formed plaques originate from laban6-1 e.g. by plaque PCR and sequencing? Can exclude the possibility that the plaques are not representing laban6-1?
375: 3.1. Subsection?
377: Rephrase, genera do not infect
381: comparison to Cellulophaga and Mycobacterium phages: Please clarify the comparison.
383: for each host? Please clarify, generally?
383: "lytically replicating LMO6 and LMO9 phages" lease reconsider the use of lytically replicating
386: Rephase the sentence, correct English
390: The discussion is not fluent, occasionally listing observations. The discussion must be improved to point out the relevance of the work.
The work should be linked more to the existing literature on Flavobacterium phages, Baltic Sea phages, microbial data collected at LMU etc. Reorgizationg might help.
394: "muminphages compared to LM08 phages". It is very difficult to follow such comparisons. "... to muminphages infecting LM06 and LMO9.
397: "that spans across genera" does this refert o bacterial or viral genera?
403: Clarify the barbaphages. Also tailed viruses infecting Flavobacterium?
408: were to various phages and bacteria within the host family. Rephrase
410: add reference to VICTOR Fig
412: VICTOR or Victor?
413-415: unclear sentence
417: Did you identify the genes for encoding packaging terminase, tail sheet protein and major capsid protein (HK97 fold) etc - the hallmarks of the tailed dsDNA bacteriophages in the order Caudovirales?
433: Host growth rate 91 min? What this mean?
437: eDNA -> clarify the meaning of this for the reader
440: "the presence of laban6-1 provided LMO6 with resistance to new infections of laban6-1 through superinfection exclusion," Have you shown this? Unclear statement
446: "do not induce all" -> do not cause virus induction in all lysogenised bacteria
507: Provide a figure legend within the Supplementary Fig. 1 where you indicate what is the strain.
508:
Table S1
Virus-name: is this proposed species name?
Edited_Lenght: genome lenght (bp)
Here you have the information about the circular nature of the molecules, please use this information in the main text.
What means "no-genes", "genes_with_conserved_hypo" and "total-hits"?
Table S2: Species Genus Family columns: Please clarify the meaning of the numbers in the cells
Table S3: remove all empty columns
Table S7: Indicate in the table which one of the strains is the original host
All supplemental tables: add analysis method/tool etc and its reference
Author Response
Nielson et al have isolated and studied a number of phages infecting Flavobacterium strains from Baltic sea and analysed their genome sequencing. I was delighted to learn that they have decided to call these phages by using lovely character names originating from famous Finnish and Swedish writers Tove Jansson and Astrid Lindgren.
Thank you, we are delighted by these names too.
Although the study has been done with scientific rigors, the text is occasionally difficult to follow and understand. The manuscript text needs to be improved, especially the materials&methods and discussion. The figure legend and table titles and table column heading are occasionally incomplete.The work is quite descriptive. It would benefit if the discussion could link the study better to the known literature. Especially, I would like to know for example that is this kind of studies done before in Baltic and or related environments. Is the host range linked to the virus morphology observed? Myoviruses typically have broader host range than siphoviruses. The host genome sequencing is not documented so that at least I could not follow has it done here or somewhere else. Data and/or references need to be provided.
Response: Thank you for your comments for the improvement of our manuscript. We have developed our discussion to include more regarding our phages and previously described phages in the Baltic Sea, we have also expanded our discussion with regards to structural viral hallmark genes and the relationship between morphology and host range. Also, we have tried to clarify out methods description, especially with regards to mapping of the sequence sample with the temperate phage and the lysogenised part of the bacteria. We have also improved how we describe the bacterial genomes and their comparisons with the references we used, both by adding that GTDBTk compares several marker genes and rewriting the text in the results and Figure 1 for clarity.
The virus taxonomy is moving forward quite fast. It would be good to use siphoviruses and myoviruses as a virus morphology in the text and forget Siphoviridae and Myoviridae (the taxonomic units), since families and morphologies does not always follow among members of the order Caudovirales. There are currently 10 families in the order Caudovirales
Response: We agree with your comment – virus (phage) taxonomy is moving fast and family genetics and morphology do not agree well. Therefore, we have already used the phrasing “siphovirus morphology” and “myovirus morphology” for our described phages. Our only mention of Siphoviridae is for a previously described phage, Flavobacterium phage Fpv5, for which we now have changed the text to siphovirus, see line 797.
Please note, all the line numbers we are referring to are the line numbers as they appear in the Microsoft Word track changes-version that we are submitting. Since the document contains tracking the numbers seem higher than they should, but we hope that they can make it easier to find where we have done the relevant revisions.
Line 2: Reconsider the title. Functionality does not necessarily describe the content of the work properly.
Genetic diversity and virus-host interactions among...
Response: Thank you for this comment, we have now changed the title to “Diversity and host interactions…” instead of functionality. We prefer to not add “Genetic” diversity, as we also show morphological diversity.
14: Family - refers to taxonomy? Group could be better
Response: Here we mean the taxonomic order of the host and since we talk about a phylum earlier in the sentence we understand if family is a confusing word to use. However, we still want to refer to the taxonomic level and have therefore changed “family” to “phylum” see line 14.
16: How many sequenced? This should be said in the abstract
Response: We have now clarified that all isolated bacteria and phages also were sequenced, see line 16.
21-24: Rephrase. Too detailed for abstract, make a conclusion.
Response: Thank you for your comment. We have now tried to make this section less detailed and more concluding, see line 21-23
53: infect members of Flavobacteriaceae
Response: We have now added “members of”, see line 251
72: Materials and Methods: Please provide subheadings for this section.
Response: Thank you for this comment, we agree that subheadings makes this section easier to follow and have added them accordingly.
78: MQ -> distilled water, correct also in other places
Response: We have changed “MQ” to “milliQ” water and not distilled (e.g. see line 274), since the water we have used comes from one of Millipores milliQ systems where the water is extensively filtered and not just distilled. The water that is fed into the milliQ system is deionized water.
124-138: How you know that you have induced only one phage? Did you plaque purified it?
Response: We have not done plaque purification of the induced phage. On the lysate that was made from the induced phages that did replicate on LMO9, we preformed TEM analyses and all visualised phages showed similar size and shape. This phage morphology was shared with the low-number of phage particles with different morphology (the temperate phage) compared to the majority of phage particles (the lytic phage) in the lytic-temperate phage mix (compare fig 6 d [only temperate phage replicated on LMO9] and e [mix of lytic and temperate phage]). From the virus sequencing, we were never able to detect any other induced phage genomes nor were we able to detect any other potential prophages in the LMO6 genome using phage finder tools (PHASTER), methods and results of this has been added see line 431 and lines 550-552. Given all these agreeing results we feel positive that we have induced one phage and that the induced phage in the experiment is the same as the sequenced phage.
147: DNA extraction of phages is here, but genome extraction of the strains is described above. It is difficult to follow what has been done. Maybe separate the phage sequences and strain sequences, or otherwise clearly indicate what has been done. Use subheadings.
Response: Thank you for your comment. We hope that our use of subheadings now has clarified ore method section. As the sequencing of phages and bacteria were done in the same way, this is combined under the same heading, as well as part of the bioinformatic analyses.
177: Please specify what is high
Response: We have added our similarity cut-off, e-value less than 0.001, and also changed the wording to clarify that we chose references based on significant similarities between the predicted amino acids, see line 439.
177: protein sequence similarity, see also below
Response: Thank you pointing this out, we have changed it accordingly, see lines 439 and 441 .
180: how many?
Response: In total, we used 37 reference genomes, which has now been clarified, see line 443.
200-207: mapping of ?? Please specify, difficult to understand
Response: We have tried to be more specific, both in the subheading of this section, as well as adding more specific descriptions in the text. What we try to convey is that we mapped reads from one of our sequenced wells/samples against the lytic phage that was assembled from that sample, as well as against the region within the bacteria that has the prophage. See lines 474-484.
230: This does not say are the bacterial genome sequences whole genomes or draft genomes?
Response: We have now added that we ran our assembled bacterial genomes through CheckM (see lines 418-419) and that they were high-quality draft genomes (see line 533).
233: "Whole genome comparison" Where is the data of the whole genome sequences?
Response: We have clarified that we do not do purely whole genome comparison, but instead rely on GTDBTk that compares the genomes with several marker genes, see line 434-435, 546. The high-quality draft genomes can be found by searching for the specified accession numbers on lines 534-535.
234: Flavobacterium genus and species names in italics, correct in all places
Response: Thank you for pointing this out, we have corrected this now, see line 259-262.
239: Please specify "the three strains" in the legend; add Acc Nos for each isolates into the figure; Is the figure based on whole genome sequence or on 16S?
Response: We have clarified the figure legend to include more of how the three strains were placed in the phylogeny with GTDBTk, see lines 554-556. We have also added the accession numbers for the reference genomes used.
242: lytic phages-> virulent phages: do you have data supporting this?
virulent phages have typically lytic life-cycle and lyse the cell in the end of its life-cycle, but can have also non-lytic exit strategy. The plaque can be seen e.g. due to the host growth retardation.
Response: Thank you for your comment. As we have been able to retrieve large quantities of viruses from each plaque (a pick of a plaque gave > 106 phage particles), we feel confident that the plaques were produced by replicating phages killing their hosts and not host growth retardation (only growth retardation would not produce novel phage particles). The plaques have also been seen to grow in size on the lawn of bacteria, indicating the bacteria that previously were there (intact bacteria that could be seen as part of the lawn) have been lysed (we cannot see them anymore, instead a larger clear zone).
Potentially, the “lytic” phages could replicate through lysogenic replication, however, we have not detected any genes indicating the ability of lysogenic replication cycles, nor seen the turbid plaque formation commonly detected for lysogenic replicating phages.
Given this, and that lytic and lysogenic replication is the most common replication cycles for Caudoviruses, we feel confident in calling our “lytic phages” lytic.
244: Are the genomes dsDNA? Circular or linear?
Response: The sequencing method we use only targets dsDNA and we have validated the circularity of the genomes by looking for identical k-mers at both ends of the genome, which we also have clarified in the text, see lines 569-570.
250: Do you mean "in three new genera and 15 new species"?
Response: Yes, this is what we mean and we have added this, see lines 576-577.
261: Muminvirus is orange
Response: Thank you for pointing this out, we have changed from “black” to “orange”.
268: with the newly isolated phages (Delete lytic and temperate); Acc. Nos for the reference genomes would be goof to have in the fig. Are these phages all isolated and cultivable or does this include also prophages?
Response: We have changed to “newly isolated phages”. The information of which reference genomes are included is in the method section (2.9, which should hopefully be easier to find since we now use subheadings), and there we indicate that we have included potential prophages found in bacterial genomes for the phylogenetic analysis, see lines 441-446.
272: Known / Predicted functions for these shared genes?
Response: We have now indicated which these genes are in the supplemental tables with either “*” or “**” at the gene names, but as the similarity between these genes are low (25-37%, see lines 600 and 659) we do not see an interest in discussing this further.
274: What is the size range of the genomes within one genus?
Response: This information is available in the supplemental table S1 (see line 570-572). The variation in the genus with most isolates is 2.5 kbp between the longest and shortest genome (average is 38.3 kbp, ± 0.6 kbp).
287: add genome lengths, ORF / gene numbers above the arrows, mark the four shared genes (see comment above); Are these linear molecules?
Response: Adding more information, such as genome length and gene numbers, would make the figure more informative but we are afraid the it would also become more difficult to read and detract from what we want to highlight (i.e. genes with functional annotations and their organization). Also, marking the four shared genes according to blastn might be confusing, since EasyFig did not identify them as sharing any similarities. EasyFig also relies on blast, but we performed a blastn analysis there instead of the blastp that we used for comparing the genes directly with blast . However, we have added a line encouraging the reader to look at the supplemental tables in order to get more details, see line 623-624. We have also pointed out that the genomes are shown linear for visualization.
290: phage cultures (you have not purified phage particles biochemically)
Response: That is correct, and we have changed the sentence, see lines 626-627.
292: "a phage with lytic replication (described above), which was the phage that was supposed to be sequenced" (delete) -> concluded that one of the genomes was the phage (give here the name of the phage).
Response: Thank you for your suggestion, we have altered the text accordingly, see line 629.
296: temperate phages -> sequences
Response: This has been changed, see line 633.
298-302: Unclear. Rephrase
Response: To clarify, this sentence has been rewritten and shortened, see line 624-627.
304: Have you used all obtained viral reads? What is NODE-12?
Response: We used all sequenced (except of low quality or with too much adapters) and mapped them against the contig of LMO6 that had the prophage region, this contig is named “NODE_12”. We have specified this, see lines 654-655.
305: The temperate phage was 43,236 nucleotides long -> The temperate phage genome was 43,236 nucleotides long. Is this genome dsDNA, circular, linear?
Response: We have added this information, see lines 657-658.
306: What these genes are? Give gene names and refer to data in a Table.
Response: We have added a reference to the supplemental tables, where this information can be found – see response to your comment regarding the same issue for the lytic phages.
310: a putative tyrosine...
Response: Thank you for pointing this out, we have added putative, see line 663.
313: Is Fig2 correct figure for this sentence? Add references as examples that what these similar phages are
Response: No, the correct figure should be Fig 3, thank you. We do discuss which these phages and prophages are in the discussion, but we have added an example of each, see lines 665-666. Here we are only reporting that VICTOR also recognises our new phage as a distinct species.
328: the phages tooticki6-1 and laban6-1
Response: We have been unable to modify our text according to this comment, since we did not understand if you want us to add “the phages tooticki6-1 and laban6-1” somewhere or remove it.
330: Correct the Figure name
Response: We have now corrected the names for the figures, see lines 683, 695.
332-334: Is the culture mixture of several viruses?
Response: The only culture that is a mixture of the cultures used for the micrographs is the one for figure 6e. The culture that underlie that micrograph is the phage culture for tooticki6-1, which was isolated from LMO6 and therefore contains the prophage, laban6-1, which was induced during the isolation of tooticki6-1.
336: Provide a high-resolution figure, now too low-resolution
Response: We will provide a figure with as high resolution as possible.
342: Mark the original host of each virus in fig 7.
Response: The original host is indicated in the name of each phage, e.g. mumin9-1 indicates that this phage was isolated on LMO9. We have now clarified this in the figure legend.
348: "on LMO6 and then harvested before new plaque assays were performed on LMO6 and LMO9."
-> on LMO6 and plaque assays were performed on LMO6 and LMO9.
Response: Thank you for this suggestion. We have now changed the sentence to make it clear that the phages first were grown on LMO6 and after that were grown on LMO6 and LMO9, see line 709.
349: plaques per ml -> PFU/mL; Correct in all places
Response: Thank you for pointing this out. We have corrected this in all relevant places, e.g. lines 711-712.
347-351: Summarize what is the results. Influence?
Response: We have summarised what information we gained from these results, namely that we got similar efficiency of plating on both LMO6 and LMO9 after passing the phages through LMO6, this is clarified on lines 710-712.
353: What mean _1, _2, _3 in each bacterial isolates? Clarify this in the figure legend.
Response: We have now clarified that this indicates which replicate the efficiency of plating represent, see lines 727-729.
359: Have you checked that the formed plaques originate from laban6-1 e.g. by plaque PCR and sequencing? Can exclude the possibility that the plaques are not representing laban6-1?
Response: We have not verified that the DNA is the same for the plaques as laban6-1, but we do have electron micrographs where we only get one type of phage for the plaques and these match the electron micrographs of the rare phage from the sample with both tooticki6-1 and laban6-1.
375: 3.1. Subsection?
Response: That seem to have been a remnant from merging our manuscript into the viruses-template and has now been removed.
377: Rephrase, genera do not infect
Response: Thank you for noticing. We agree and have now rewritten to indicate that it’s the members of the genera that infect and nothing else.
381: comparison to Cellulophaga and Mycobacterium phages: Please clarify the comparison.
Response: We want to point out that the genome sizes of our phages are more limited than the ranges of genome sizes in these systems, and we have modified accordingly see lines 763-765.
383: for each host? Please clarify, generally?
Response: We have clarified that we discuss the diversity of phages that infect the strains LMO6 and LMO9, see line 765-768.
383: "lytically replicating LMO6 and LMO9 phages" lease reconsider the use of lytically replicating
Response: “lytically” has been changed to “lytic replication”, see lines 766-767.
386: Rephase the sentence, correct English
Response: We have rewritten and reordered the sentence and moved the parentheses to the end in order to make the message we want to convey clearer, see lines 769-773.
390: The discussion is not fluent, occasionally listing observations. The discussion must be improved to point out the relevance of the work.
The work should be linked more to the existing literature on Flavobacterium phages, Baltic Sea phages, microbial data collected at LMU etc. Reorgizationg might help.
Response: We have worked on summarizing and concluding our results more as well as putting them in a context of previous research, e.g. see lines 784-790 regarding the presence of muminphages at LMO, or lines 795-803 regarding the similarity between our phages and the references we used – where we do not find similarities in spite of the fact that the reference phages also infect Bacteriodetes, or lines 1119-1122 where we discuss the efficiency of plating depending on last host infected. We hope that this makes the discussion easier to follow, and that the benefits with this system, which we believe is a great resource with its diversity of phages and their varied interactions with their hosts, is properly discussed.
394: "muminphages compared to LM08 phages". It is very difficult to follow such comparisons. "... to muminphages infecting LM06 and LMO9.
Response: Thank you for pointing this out. Instead of comparing muminphages to “LMO8 phages” we are instead using “pippi- or tantphages” and have clarified which hosts are the original hosts for the different groups, e.g. see lines 777-778.
397: "that spans across genera" does this refert o bacterial or viral genera?
Response: We thank you for pointing this out, this should of course indicate the phages diversity with regards to genera and morphologies and we have modified on line 782.
403: Clarify the barbaphages. Also tailed viruses infecting Flavobacterium?
Response: Thank you for pointing this out. The barbaphages are infecting Gammaproteobacteria, but were isolated at the same time and place as the phages in this study. We have added information describing this, see lines 785-788.
408: were to various phages and bacteria within the host family. Rephrase
Response: We have clarified that the hits were to phages as well as bacteria that belong to the same family as the phages’ hosts, see lines 793-795.
410: add reference to VICTOR Fig
Response: We have added a reference, thank you for noticing the lack of it.
412: VICTOR or Victor?
Response: Thank you for pointing this out, it should be VICTOR which is now changed.
413-415: unclear sentence
Response: We split the sentence in two to be able to first convey that we managed to annotate functions, and then that the genes are not similar across genus, but functions are. See lines 804-806.
417: Did you identify the genes for encoding packaging terminase, tail sheet protein and major capsid protein (HK97 fold) etc - the hallmarks of the tailed dsDNA bacteriophages in the order Caudovirales?
Response: We did identify terminases and capsid proteins in some but not all phages and no tail sheath protein were recognized in any of them. It is not uncommon that these genes are not recognized in novel aquatic phages. We have added a couple of lines describing this, see lines 1019-1025.
433: Host growth rate 91 min? What this mean?
Response: We apologise, it should be generation time and not growth rate, see lines 1038-1039.
437: eDNA -> clarify the meaning of this for the reader
Response: We have now clarified that we mean extracellular DNA, see line 1042.
440: "the presence of laban6-1 provided LMO6 with resistance to new infections of laban6-1 through superinfection exclusion," Have you shown this? Unclear statement
Response: We have not shown that laban6-1 as a prophage stopped infections through superinfection exclusion, which is why we also propose that it could be homoimmunity, but we have shown that LMO6 with the prophage did not produce plaques when plaque assays with laban6-1 was performed. This has been clarified on lines 1045-1049.
446: "do not induce all" -> do not cause virus induction in all lysogenised bacteria
Response: We have rephrased this according to your suggestion which we thank you for.
507: Provide a figure legend within the Supplementary Fig. 1 where you indicate what is the strain.
Response: We have added a figure legend in Figure S1 and also clarified which strain that was used in the information about the supplemental files, see line 1141.
508:
Table S1
Virus-name: is this proposed species name?
Response: Yes, this is the proposed species name and it has been clarified in the name of the column. Overall, we have improved the names of the columns throughout the supplemental tables.
Edited_Lenght: genome lenght (bp)
Response We have changed to “genome length (bp)” now.
Here you have the information about the circular nature of the molecules, please use this information in the main text.
Response: This has been added in the main text, see lines 569-570 and 657-658..
What means "no-genes", "genes_with_conserved_hypo" and "total-hits"?
Response: We have rephrased this to better indicate what we mean, i.e. number of predicted genes, number of genes with a functional annotation, number of genes annotated as hypothetical genes and total number of genes with hits to ncbi or pfam.
Table S2: Species Genus Family columns: Please clarify the meaning of the numbers in the cells
Response: We have added an explanation that the numbers indicate which group that VICTOR has assigned them as in the Table S2 heading.
Table S3: remove all empty columns
Response: We have added the information that were missing in the empty columns in Table S4-S7 (previously Table S3-S6). Thank you for pointing this out!
Table S7: Indicate in the table which one of the strains is the original host
Response: This has been indicated with shading the relevant cells with a light grey colour, see Table S8.
All supplemental tables: add analysis method/tool etc and its reference
Response: Thank you for pointing this out, we have added the missing analysis tools and added references to each within the excel-sheets, we hope this is satisfactory.
